# REG3A/REG3B promotes acinar to ductal metaplasia through binding to EXTL3 and activating the RAS-RAF-MEK-ERK signaling pathway

Huairong Zhang[1,2,3,7], Andrea Liliam Gomez Corredor[2,7], Julia Messina-Pacheco [2,7], Qing Li[4], George Zogopoulos[5], Nancy Kaddour[6], Yifan Wang[5], Bing-yin Shi[3], Alex Gregorieff [2], Jun-li Liu [6✉] & Zu-hua Gao [2✉]

Persistent acinar to ductal metaplasia (ADM) is a recently recognized precursor of pancreatic ductal adenocarcinoma (PDAC). Here we show that the ADM area of human pancreas tissue adjacent to PDAC expresses significantly higher levels of regenerating protein 3A (REG3A). Exogenous REG3A and its mouse homolog REG3B induce ADM in the 3D culture of primary human and murine acinar cells, respectively. Both *Reg3b* transgenic mice and REG3B-treated mice with caerulein-induced pancreatitis develop and sustain ADM. Two out of five *Reg3b* transgenic mice with caerulein-induced pancreatitis show progression from ADM to pancreatic intraepithelial neoplasia (PanIN). Both in vitro and in vivo ADM models demonstrate activation of the RAS-RAF-MEK-ERK signaling pathway. Exostosin-like glycosyltransferase 3 (EXTL3) functions as the receptor for REG3B and mediates the activation of downstream signaling proteins. Our data indicates that REG3A/REG3B promotes persistent ADM through binding to EXTL3 and activating the RAS-RAF-MEK-ERK signaling pathway. Targeting REG3A/REG3B, its receptor EXTL3, or other downstream molecules could interrupt the ADM process and prevent early PDAC carcinogenesis.

[1] Department of Endocrinology and Metabolism, Shanghai General Hospital, Shanghai Jiao Tong University School of Medicine, Shanghai, China. [2] Department of Pathology, McGill University and the Research Institute of McGill University Health Centre, Montreal, QC, Canada. [3] Department of Endocrinology, The First Affiliated Hospital of Xi'an Jiaotong University, Xi'an, Shaanxi, China. [4] Human Oncology and Pathogenesis Program, Memorial Sloan Kettering Cancer Center (MSKCC), New York, NY, USA. [5] Department of Surgery, McGill University and the Research Institute of McGill University Health Centre, Montreal, QC, Canada. [6] Department of Medicine, McGill University and the Research Institute of McGill University Health Centre, Montreal, QC, Canada. [7] These authors contributed equally: Huairong Zhang, Andrea Liliam Gomez Corredor, Julia Messina-Pacheco. ✉email: jun-li.liu@mcgill.ca; zu-hua.gao@mcgill.ca

Pancreatic ductal adenocarcinoma (PDAC) remains one of the most lethal human cancers with a 5-year survival rate <5%. The poor prognosis of PDAC has been attributed to multiple factors including late diagnosis, the lack of sensitive and specific biomarkers to detect PDAC, and the lack of effective measures to prevent its development and interrupt its progression[1]. Three precursor lesions of PDAC have been identified: intraductal papillary mucinous neoplasm, pancreatic intraepithelial neoplasia (PanIN), and mucinous cystic neoplasm[2]. PanIN is by far the most well-characterized precursor lesion with a histological step-wise progression from low-grade to high-grade PanINs and then to invasive adenocarcinoma, which correspond molecularly to a series of genetic events including the activation of the *Kras* oncogene and the inactivation of the tumor suppressor genes *Cdk2na*, *Tp53*, and transcriptional factor *Smad4/Dpc4*[3].

It has been well documented that PanINs can arise from pancreatic acini undergoing acinar to ductal metaplasia (ADM)[4,5]. The normal pancreatic acinus is composed of wedge-shaped cells containing zymogen granules arranged around a small lumen. Acinar differentiation contributes to pancreatic tissue homeostasis and the function of the exocrine pancreas. Histologically, ADM is characterized by acinar cells losing zymogen granules and adopting the appearance of duct-like structures. Loss of acinar markers such as *Mist1*, *Amylase*, *Carboxypeptidase*, *Elastase*, and gain of ductal markers such as *Cytokeratin 19* (*Ck19*), *Mucin 1*, *Duodenal homeobox 1* (*Pdx1*), and *Sry-related high-mobility group box 9* (*Sox9*) are hallmarks of ADM[6–9]. In patients with pancreatitis or other forms of pancreatic injury, ADM is transient and reversible[10,11]. However, persistent ADM may progress to PanIN, and finally to an invasive tumor[9,12–14]. It is yet unclear what molecular mechanism drives the formation and malignant progression of ADM.

The family of regenerating (REG) proteins is a group of C-type lectin-like proteins discovered in patients with pancreatitis and during pancreatic islet regeneration after injury. Five REG family members including REG1A, REG1B, REG3A, REG3G, and REG4 have been identified in humans[15]. REG3A (homolog of mouse REG3B), also known as pancreatitis-associated protein (PAP) due to its upregulation in response to inflammatory stimulants, is generally expressed at low level in normal pancreas[16,17]. REG3A has been described as a proliferating factor following liver and skin injuries as well as a driver of pancreatic cancer cell growth through JAK2/STAT3 signaling pathways in response to interleukin-6[16,18]. Other members of the REG proteins family, i.e., REG1A and REG4, have been shown to be overexpressed in PDAC patients and to accelerate cell proliferation and tumor growth in vitro[17,19,20]. We previously reported that ADM can be induced by adding recombinant REG3B in the 3D culture of acinar cells in the context of wild type *Kras*[21]. In the current study, we demonstrated the involvement of REG3A/REG3B in the ADM process using several models including: human pancreas tissue from patients with PDAC, in vitro cultures of primary human and murine pancreatic acinar cells, an in vivo transgenic mouse model with caerulein-induced pancreatitis, and recombinant REG3B-treated wild-type mice with caerulein-induced pancreatitis. We also identified a candidate receptor for REG3B on the acinar cell membrane, and unveiled the downstream signaling pathway that leads to ADM formation and maintenance.

## Results

**REG3A is overexpressed in the ADM area adjacent to PDAC in human pancreas.** To determine the expression pattern of REG3A, we performed immunohistochemical staining on human pancreatic tissue sections that contained areas of transition from normal acini to ADM and to PDAC (Fig. 1a with enlarged view of

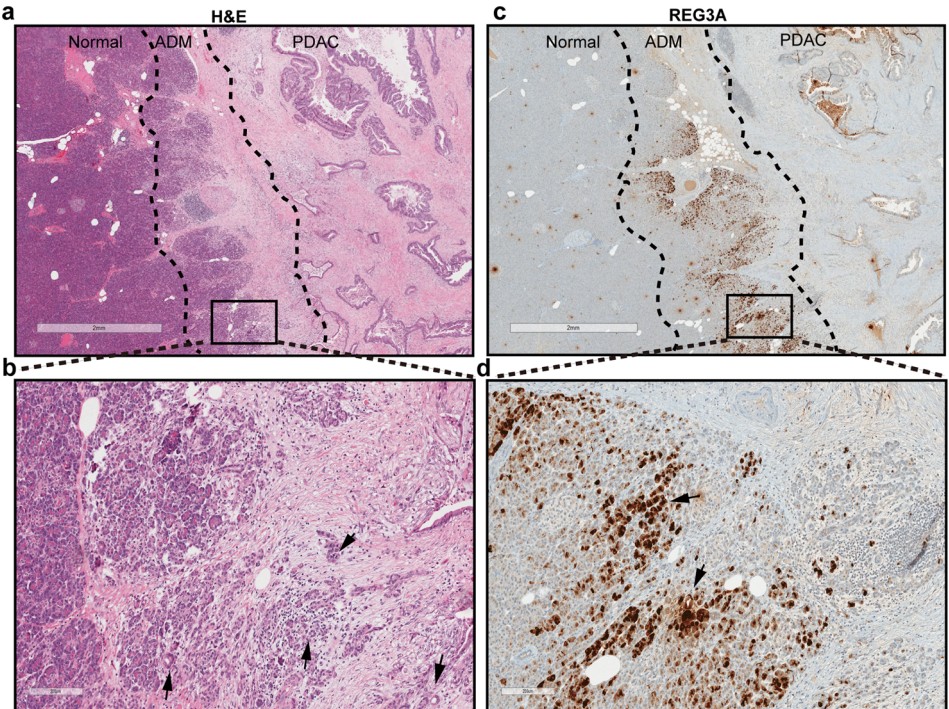

**Fig. 1 REG3A is overexpressed in the ADM area adjacent to human PDAC. a** H&E staining shows histological evidence of transformation from normal acini to ADM and to PDAC. Magnification, ×10; Scale bar: 2 mm. **b** Enlarged view of ADM area in **a**. Magnification, ×100; Scale bar: 200 μm. Black arrows indicate typical ADM circular structures. **c** Single antibody immunohistochemical staining shows intense REG3A expression the ADM zone. Magnification, ×10; Scale bar: 2 mm. **d** Enlarged view of ADM area in **c**. Black arrows indicate typical ADM stained strongly with REG3A. Magnification, ×100; Scale bar: 200 μm.

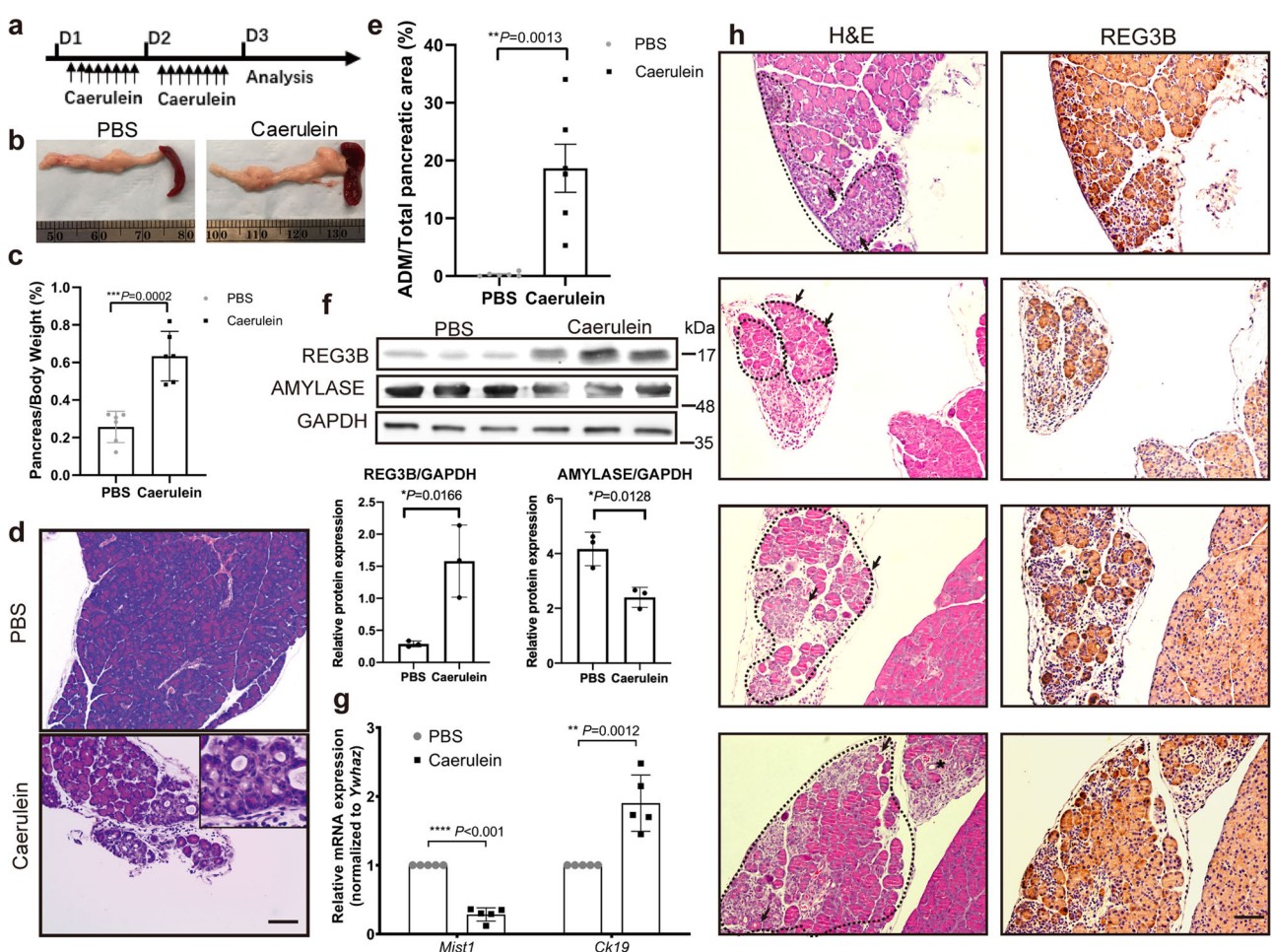

**Fig. 2 REG3B is elevated in caerulein-induced ADM in mice. a** Scheme of caerulein injection to induce acute pancreatitis. Acute pancreatitis was induced by intraperitoneal (i.p.) injection with caerulein dissolved in PBS at a dose of 50 μg/kg at hourly intervals eight times daily, for two consecutive days. PBS injection alone served as control. **b** The size of pancreas of PBS-treated mice appear smaller than those of caerulein-treated mice ($n = 6$, student's $t$-test). **c** Percentage of pancreas/body weight of caerulein-treated mice versus PBS-treated mice ($n = 6$, student's $t$-test). **d** Abundant ADM formation in caerulein-treated mice (H&E, 40×, Scale bar: 500 μm). **e** Difference in the extent of ADM between PBS-treated mice and caerulein-treated mice, as measured by percentage of ADM area in total pancreatic area on H&E stained tissue sections ($n = 6$, student's $t$-test). **f** Western blot analysis with quantifications showed higher REG3B protein level and lower AMYLASE protein level in the pancreatic tissue of caerulein-treated mice than that of PBS-treated mice ($n = 6$, student's $t$-test). GAPDH served as a loading control. **g** RT-qPCR analysis showed higher $Ck19$ and lower $Mist1$ mRNA levels in the pancreatic tissue of caerulein-treated mice than that of PBS-treated mice ($n = 5$, student's $t$-test). **h** H&E staining of caerulein-induced ADM in mice, with corresponding REG3B IHC staining. ADM areas were highlighted by dashed lines, with arrows indicating naïve ADM and asterisks indicating relatively mature ADM. Scale bar: 200 μm. Data are represented as means ± SD; *$P < 0.05$; **$P < 0.01$; ***$P < 0.001$; Non-significant (n.s.) if $P > 0.05$.

ADM in Fig. 1b). There was a higher expression of REG3A in the ADM area than the normal, PanIN and PDAC areas (Fig. 1c with enlarged view in Fig. 1d). We then performed REG3A/CK19 double staining on pancreatic tissue sections of eight patients (Supplementary Fig. 1a–p). Immature ADM (indicated by black arrow in Supplementary Fig. 1q) showed no or very weak expression of CK19. As the ADM matures and develops a clearer duct-like morphology (indicated by asterisk in Supplementary Fig. 1q), there is a gradual increase in CK19 expression. Colocalization of CK19 and REG3A was well displayed in the enlarged view of mature ADM (Supplementary Fig. 1q). Quantification analysis showed that 99% of the ADM area stained positively for REG3A, whereas normal acini had minimal (4%) REG3A positivity (Supplementary Fig. 1r). REG3A expression was reduced in PanIN and PDAC, compared with ADM. We then measured the level of $Reg3a$ mRNA in micro-dissected areas of normal pancreatic tissue with matched ADM zone by RT-qPCR. The level of $Reg3a$ mRNA in ADM area was nearly 2-fold higher than in normal pancreatic tissue (Supplementary Fig. 2a). This data suggests that REG3A protein and mRNA are highly expressed in human ADM.

**REG3B is elevated in caerulein-induced pancreatitis and maintains prolonged ADM in mice.** To determine whether REG3B is involved in the ADM process in vivo, we challenged wild type C57BL/6J mice (WT) with caerulein to induce pancreatitis (Fig. 2a), which is a reliable and widely used mouse model for ADM induction[22]. At day 3, caerulein-treated mice had a larger pancreas-to-body weight ratio than the control group (Fig. 2b, c). Histologically, the exocrine compartment of caerulein-treated mice displayed areas of ADM, whereas the control group (PBS-injected mice) did not (Fig. 2d). Quantification analysis showed a statistically significant increase in ADM events in the caerulein-treated WT mice compared to PBS-treated mice (Fig. 2e). Western blot analysis showed an increase of REG3B protein with a concomitant reduction of AMYLASE in caerulein-treated WT mice (Fig. 2f and Supplementary Fig. 4a).

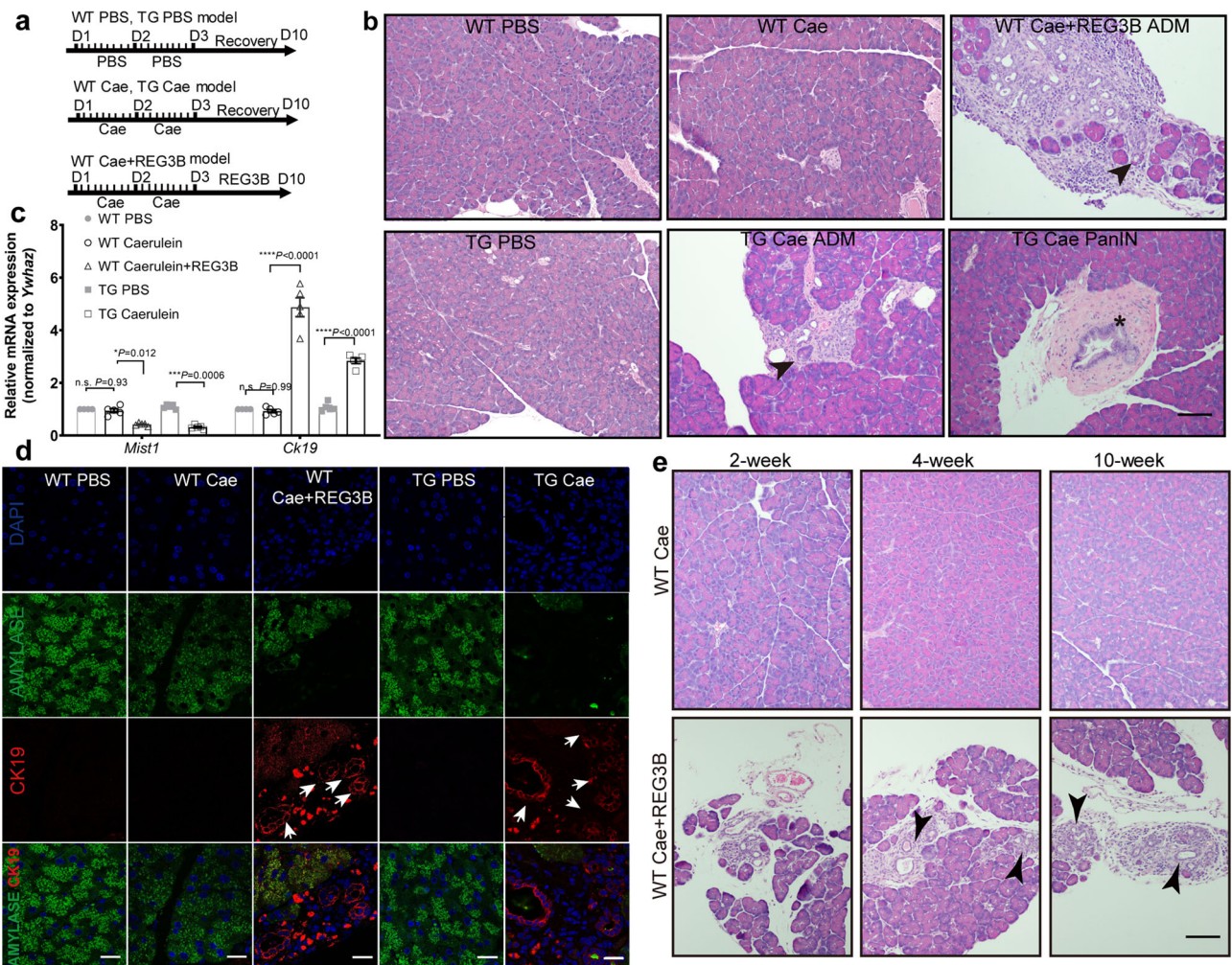

**Fig. 3 Enhanced Reg3B maintains prolonged ADM in mice. a** Scheme of intraperitoneal injection of different reagents in the five experimental groups. **b** At day 10 (D10), no ADM phenotype was observed in WT mice injected with caerulein. WT mice injected with both recombinant mouse REG3B protein and caerulein, and the REG3B TG mice injected with caerulein showed persistent ADM as indicated by the ductal morphology (black arrow heads) in H&E staining. Focal PanIN (marked by black asterisk) was observed in two out five REG3B TG mice injected with caerulein. Scale bar: 200 μm. **c** WT mice injected with recombinant REG3B and caerulein and the REG3B TG mice injected with caerulein demonstrate higher levels of *Ck19* and lower levels of *Mist1* mRNA. (*n* = 5, one-way ANOVA). **d** Immunofluorescence staining shows gain of CK19 expression (red) and loss of AMYLASE (green) in the ADM area of WT mice injected with recombinant REG3B and caerulein and in the *Reg3b* TG mice injected with caerulein. Scale bars: 20 μm. White arrows indicate ADM. **e** Persistence of ADM in WT mice injected with recombinant REG3B 2, 4, and 10 weeks after caerulein injection. (*n* = 5, magnification ×200, scale bar: 200 μm). Values in graphs are means ± SD, *$P < 0.05$; **$P < 0.01$; ***$P < 0.001$. Non-significant (n.s.) if $P > 0.05$. *n* = 5 per group, WT wild type, TG REG3B transgenic mice, Cae caerulein.

Gain of ductal marker Ck19 mRNA and loss of acinar marker Mist1 mRNA in caerulein-treated mice was demonstrated by RT-qPCR (Fig. 2g). Immunohistochemically, REG3B was highly enriched in ADM areas, particularly in naïve ADM (Fig. 2h). Therefore, ADM can be induced 1 day after the final caerulein injection, and correlates with REG3B overexpression in the pancreas tissue. However, at day 10, no ADM phenotype was observed in WT mice or PBS control groups (Fig. 3b, WT PBS and WT Cae groups). This suggests that the caerulein-induced ADM in WT mice detected at day 3 (Fig. 2e) was transient, and the pancreas almost completely recovers after 1 week.

To verify whether *Reg3b* promotes and maintains ADM in vivo, we used a gain-of-function *Reg3b* transgenic (TG) mouse model[23]. Western blot analysis confirmed a 2.7-fold higher REG3B expression level in the pancreas of TG mice than that of WT mice (Supplementary Fig. 2b). We subjected TG and WT mice to caerulein-induced pancreatitis, along with a REG3B group (WT injected with caerulein and recombinant REG3B

protein) and two control groups (WT and TG injected with PBS) (Fig. 3a). In contrast to the disappearance of ADM at day 10 in caerulein-treated WT mice (Fig. 3b), both the WT Cae injected with recombinant REG3B and the TG Cae mice showed persistent ADM one week after the final caerulein injection (Fig. 3b, WT Cae+REG3B and TG Cae groups). Furthermore, focal PanIN was observed in two out of five REG3B TG mice injected with caerulein. Reduction in the mRNA expression of *Mist1* and increase in *Ck19* were also found in both WT Cae +REG3B and TG Cae groups compared to the control groups (WT PBS, WT Cae, and TG PBS) 1 week after the final caerulein injection (Fig. 3c). This was further validated by immunofluorescence staining of CK19 and AMYLASE. WT Cae+REG3B and TG Cae mice showed a higher expression of CK19 and a lower expression of AMYLASE in the ADM area than the control groups (WT PBS, WT Cae, and TG PBS) (Fig. 3d). To further explore the capacity of ectopic delivery of REG3B on maintaining ADM, we analyzed the histomorphology of WT Cae+REG3B and

WT Cae groups at 2, 4, and 10 weeks after the final caerulein injection. Strikingly, scattered ADM lesions were observed at 2, 4, and 10 weeks after the final injection of caerulein (Fig. 3e and quantification data in Supplementary Fig. 2c). This data clearly demonstrates that REG3B treatment was able to maintain ADM and even promote ADM to PanIN in some TG Cae mice.

**Exogenous REG3A/REG3B induces ADM in human and mouse primary acinar cells in vitro.** To directly visualize the effect of REG3B on ADM formation and facilitate molecular pathway analysis, we established an in vitro 3D culture model of ADM for both human and murine acinar cells using TGFα-induced ADM as a positive control[24]. Recombinant REG3A or REG3B protein induced ADM in human and mouse primary acinar cells, respectively, as evidenced by spherical-like ductal morphology comparable with the TGFα-treated group (Fig. 4a). Quantification analysis revealed a 3-fold to 4-fold increase in

ADM events in REG3A-treated or REG3B-treated primary acinar cells compared with untreated acini (Fig. 4b). Furthermore, increased expression of CK19 and decreased expression of AMYLASE in REG3A-treated or REG3B-treated primary acinar cells was revealed by immunofluorescence staining (Fig. 4c). Consistently, in primary murine acinar cells, we found a higher level of mRNA expression of ductal-specific markers Ck19 and Nestin, and a lower level of mRNA expression of acinar-specific markers Mist1, Ptf1a, and Cpa by RT-qPCR analysis (Fig. 4d). This finding was also observed in human primary acinar cells (Fig. 4d). Therefore, both REG3A and REG3B were able to induce ADM in the 3D culture model.

**REG3B induces ADM through the RAS-RAF-MEK-ERK signaling pathway.** KRAS hyperactivity, mitogen-activated protein kinase (MAPK) signaling, and inflammatory signaling pathways have all been described to different extents as contributing factors on

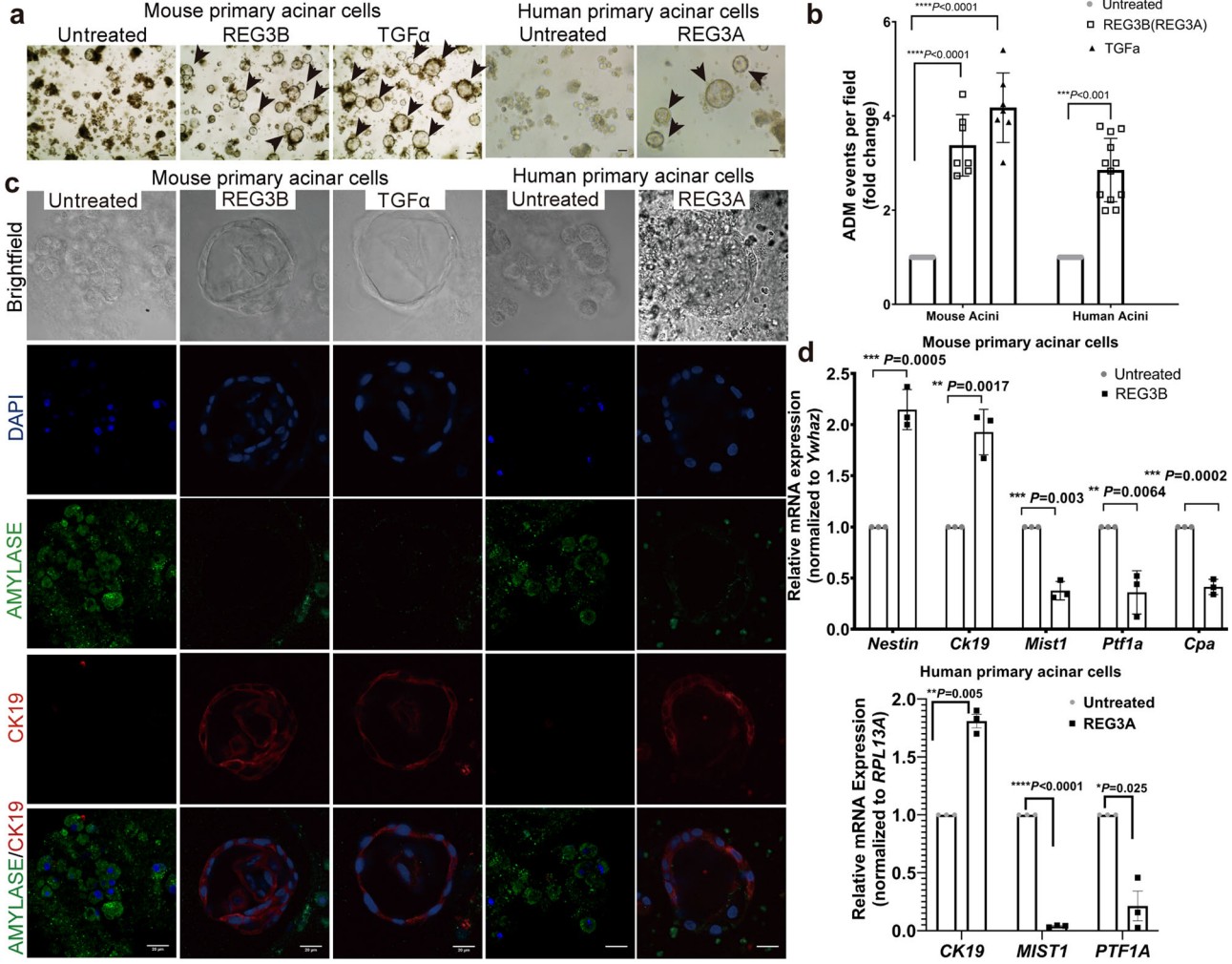

**Fig. 4 Exogenous REG3A/REG3B induces ADM in human and mouse primary acinar cells in vitro. a** Bright field images showing an increase in ADM events (depicted by black arrows) in cultured mouse primary acinar cells after 5 days of REG3B treatment and in cultured human primary acinar cells after 5 days of REG3A treatment (Magnification, ×200). TGFα-treated mouse primary acinar cells served as a positive control. **b** Bar graph showing increase in ADM quantity in 3D culture of mouse and human primary acinar cells during the 5-day REG3B or REG3A or TGFα treatment (n = 3, 15 fields each group, one-way ANOVA and student's t-test). **c** Bright field images in the upper row showing ADM events in cultured mouse primary acinar cells after 5 days of REG3B treatment and in cultured human primary acinar cells after 5 days of REG3A treatment (magnification, ×630). Lower four rows are corresponding confocal immunofluorescence images showing a decrease in AMYLASE protein expression (green) and an increase in CK19 (red) protein in REG3B-induced, REG3A-induced, or TGFα-induced ADM (magnification, ×630. Scale bars: 20 μm). **d** RT-qPCR analysis showed a decrease in acinar-specific mRNA (Ptf1a, Cpa, and Mist1) and an increase in duct-specific mRNA (Ck19 and Nestin) in mouse and human primary acinar cells after 48 h of REG3B or REG3A treatment. (n = 3 per group, student's t-test). Values are represented as mean ± standard deviation. *P < 0.05, **P < 0.01, ***P < 0.001. Non-significant (n.s.) if P > 0.05.

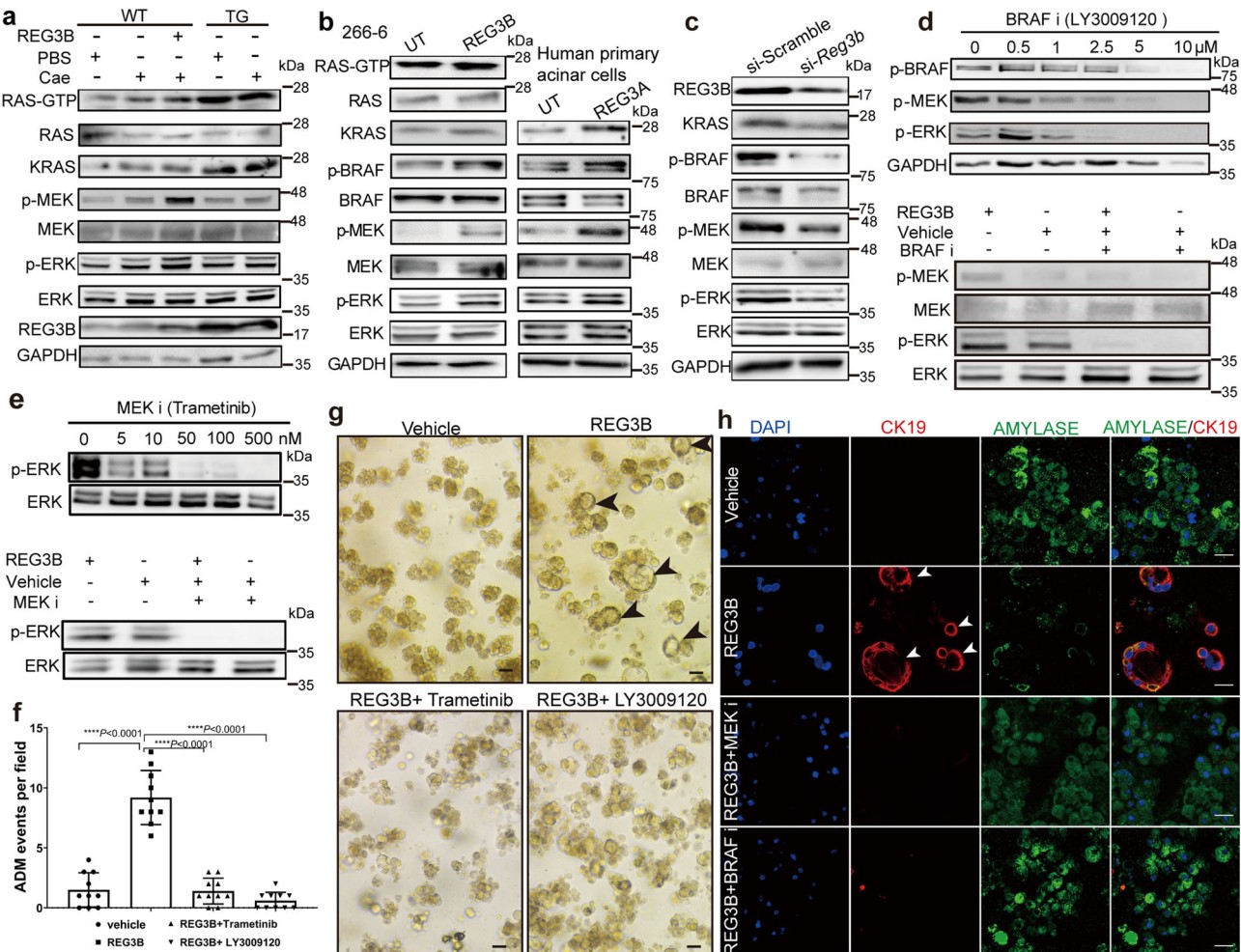

**Fig. 5 REG3B induces ADM through the RAS-RAF-MEK-ERK signaling pathway. a** Western blot showing increased expression of p-ERK, p-MEK, p-BRAF, KRAS, and active RAS in the pancreatic tissue of REG3B-treated WT mice with caerulein-induced pancreatitis (WT cae+REG3B) and a moderate increase in TG mice with caerulein-induced pancreatitis (TG cae) ($n = 3$, one-way ANOVA). **b** Western blots demonstrating increased expression of p-ERK, p-MEK, p-BRAF, KRAS, and active RAS in cultured human primary acinar cells and mouse acinar cell line 266-6 after REG3A or REG3B treatment respectively, for 48 h ($n = 3$, student's t-test). **c** Western blot showing the reduced expression of phosphorylated ERK, MEK, BRAF, and total KRAS in the 266-6 cell line after *Reg3b* gene knockdown by siRNA ($n = 3$, student t-test). **d** In the upper panel, LY3009120 inhibited ERK, MEK, BRAF phosphorylation in a dose-dependent manner in 266-6 cell line. In the lower panel, LY3009120 (5 µM) blocked REG3B-induced MEK and ERK phosphorylation ($n = 3$, one-way ANOVA). **e** Upper panel, Trametinib efficiently attenuated ERK phosphorylation in a dose-dependent manner in the AR42J cell line ($n = 3$, one-way ANOVA). Lower panel, Trametinib (100 nM) blocked REG3B-induced ERK phosphorylation. **f, g** Bright field images (**g**) with quantification (**f**) show that Trametinib (100 nM) and LY3009120 (5 µM) treatment hindered REG3B-induced ADM in 3D cultures of mouse primary acinar cells (200×, $n = 3$, one-way ANOVA). **h** Confocal microscopy shows that Trametinib (100 nM) and LY3009120 (5 µM) treatment reduced CK19 protein expression (red) and increased AMYLASE protein expression (green) in 3D culture of mouse primary acinar cells (Scale bar: 20 µm). Data are represented as means ± SD, $n = 3$. *$P < 0.05$, **$P < 0.01$, ***$P < 0.001$. Non-significant (n.s.) if $P > 0.05$.

ADM[25–27]. To elucidate the molecular mechanism of REG3B-mediated ADM, we first detected the activation of relevant signaling pathway proteins by western blot in both in vivo and in vitro models. We then validated our findings using *Reg3b* gene knockdown and specific protein inhibitors. Firstly, in in vivo experiments, we found an increase in active RAS (RAS-GTP), KRAS, p-MEK, and p-ERK in the pancreatic tissue of WT Cae+REG3B mice, TG PBS mice and TG Cae mice in comparison with WT PBS mice (Fig. 5a with quantification data in Supplementary Fig. 2b). This finding was confirmed in vitro in the 266-6 cell line treated with REG3B and in human primary acinar cells treated with REG3A, demonstrated by an increase in the expression of active RAS (RAS-GTP), p-BRAF, p-MEK, and p-ERK (active RAS was not tested in human primary acinar cells) (Fig. 5b with quantification data in Supplementary Figs. 2d, e, 4c and 5a). Secondly, siRNA knockdown of the *Reg3b*

gene dramatically reduced the expression of phosphorylated BRAF, MEK, ERK, and total KRAS in the 266-6 acinar cell line (Fig. 5c with quantification data in Supplementary Figs. 2f and Supplementary Fig. 5b). Thirdly, both LY300912, a RAF inhibitor, and Trametinib, a MEK inhibitor, inhibited BRAF, MEK, and ERK phosphorylation in a dose dependent manner and attenuated the effects of REG3B on acinar cell lines (Fig. 5d, e with quantification data in Supplementary Figs. 2g–h and 5c, d). Finally, LY3009120 or Trametinib treatment was able to completely block ADM induced by REG3B in the 3D culture model, as shown in bright field images of the morphology (Fig. 5f, g) and validated by the loss of ductal-specific CK19 and the increase of acinar-specific AMYLASE protein expression in confocal immunofluorescence images (Fig. 5h). Baseline information of AMYLASE and CK19 staining in untreated mouse primary acinar cells was given (Supplementary Fig. 2i). Collectively, our results

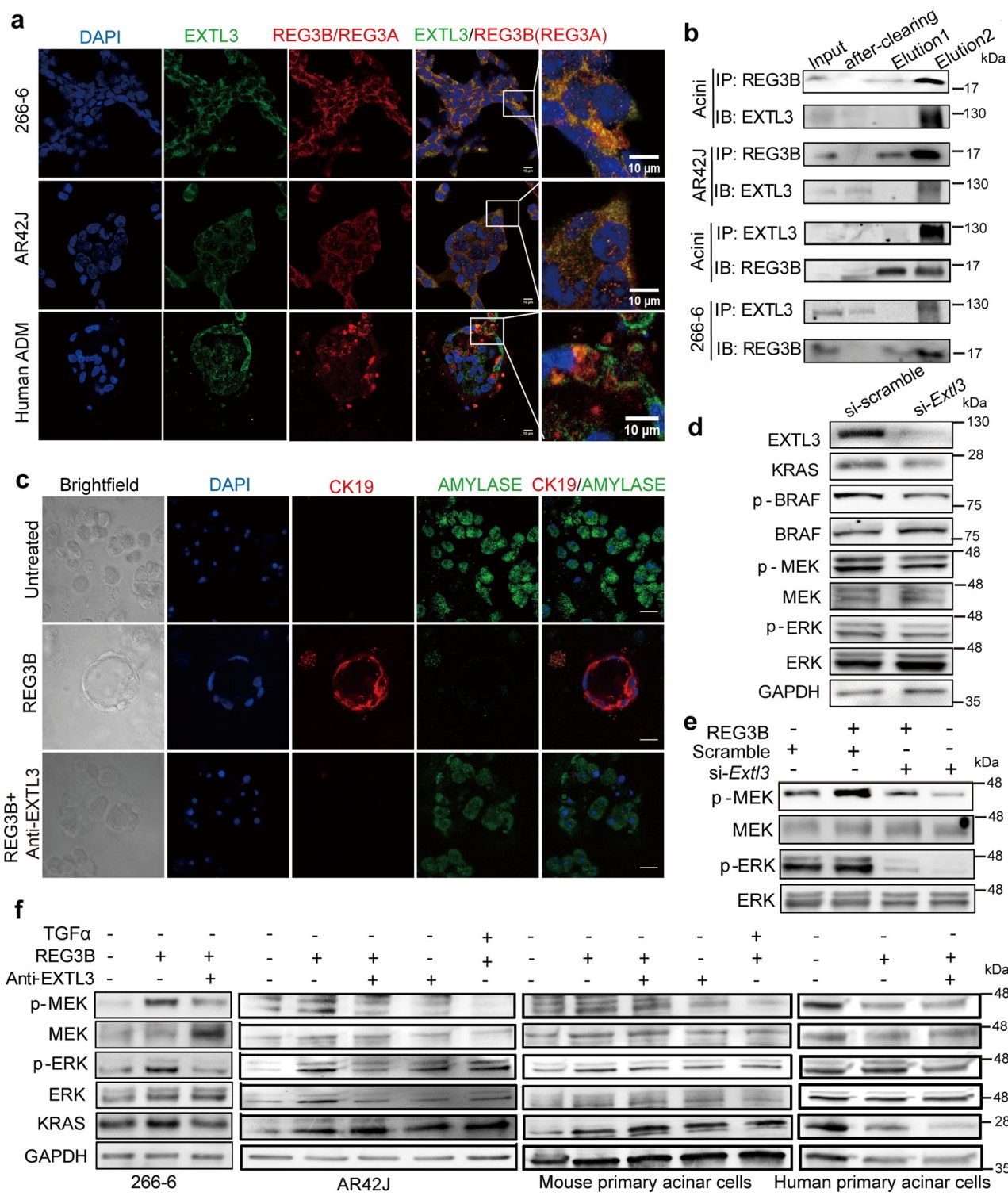

demonstrate that REG3B promotes ADM through the RAS-RAF-MEK-ERK signaling pathway.

**EXTL3 functions as a receptor for REG3B in ADM.** To find the entry point for REG3B to activate the RAS-RAF-MEK-ERK signaling pathway, we sought to identify its potential binding protein on the acinar cell membrane. It has been shown that upon REG1 or REG3A stimulation, EXTL3 activated the PI3K pathway in keratinocytes and B cells[18,28,29]. Therefore, we first counterstained REG3B and EXTL3 by dual immunofluorescence in 2D culture of REG3B-treated acinar cell lines and in 3D culture of REG3A-induced ADM in primary human acinar cells in vitro. Both the REG3B-treated acinar cell lines and the REG3A-induced ADM in primary human acinar cells showed strong co-localization of REG3B and EXTL3 (Fig. 6a). The interaction between REG3B and EXTL3 in acinar cells was further demonstrated by the detection of EXTL3 by co-immunoprecipitation with REG3B antibody in mouse primary acinar cells and AR42J acinar cell line. In 266-6 and mouse primary acinar cells, we were able to pull down REG3B with EXTL3 antibody as well (Fig. 6b and Supplementary Fig. 6a). Silencing the EXTL3 receptor with a

**Fig. 6 EXTL3 functions as a receptor for REG3B in ADM. a** Co-localization of EXTL3 (green) with human REG3A (red) or rodent REG3B (red) in ADM zones derived from human primary acinar cells in 3D culture, and AR42J and 266-6 cell lines in 2D culture by immunofluorescence microscopy. (magnification: ×630, Scale bars: 10 μm). **b** Co-immunoprecipitation of REG3B and EXTL3 in mouse primary acinar cells, rat AR42J and mouse 266-6 acinar cell lines. Lysate-bead/antibody conjugate mixture was eluted with sample buffer without DTT for 10 min at 50 °C (elution 1). Sample buffer with DTT (100 mM) was added to the pelleted beads from elution 1 and boiled for 5 min (elution 2). **c** Confocal immunofluorescence microscopy showed that EXTL3 monoclonal antibody treatment effectively blocks REG3B-induced ADM as indicated by increased expression of the acinar marker AMYLASE and decreased expression of the ductal marker CK19. (magnification: ×630, Scale bar: 20 μm). **d–f** *Extl3* siRNA or neutralizing antibody treatment reduced the protein expression of KRAS and phosphorylated ERK, MEK, and BRAF in the presence or absence of REG3B for 48 h. **d** Western blotting analysis of the knockdown of *Extl3* by siRNA (20 nM) in 266-6 cell line, **e** knockdown of *Extl3* by siRNA (20 nM) in the context of the 266-6 cell line treated with REG3B for 48 h, **f** Western blotting analysis of EXTL3 neutralizing antibody treatment (2 μg/ml) in 266-6 and AR42J cell lines and mouse primary acinar cells treated with or without REG3B for 48 h. Human primary acinar cells were treated for 30 min. TGFα treatment serves as a positive control for ADM induction.

neutralizing antibody completely abolished REG3B-induced ADM, as demonstrated by the loss of ductal morphology, and a very low expression of CK19 and high expression of AMYLASE in the 3D culture model (Fig. 6c with quantification data in Supplementary Fig. 2j). REG3B-induced ADM was impaired by blocking EXTL3 with a neutralizing antibody (Supplementary Fig. 3a). Taken together, these findings suggested that EXTL3 was not only a receptor for REG3B in acinar cells, but also facilitated ADM formation induced by REG3B. To confirm that EXTL3 is involved in regulating downstream RAS signaling molecules, we examined the effect of silencing *Extl3* by siRNA or EXTL3 neutralizing antibody on REG3B-induced RAS-RAF-MEK-ERK signaling pathway in the 266-6 acinar cell line (one out of three constructs of si-EXTL3 worked). We observed a slight reduction in the expression of total KRAS and phosphorylated BRAF, MEK, and ERK proteins after silencing *Extl3* in the 266-6 acinar cells (Fig. 6d with quantification data in Supplementary Figs. 3b and 6b). In the context of REG3B-treated 266-6 acinar cells, REG3B-induced phosphorylated MEK and ERK and total KRAS expression was successfully inhibited upon silencing *Extl3* expression by siRNA (Fig. 6e with quantification data in Supplementary Fig. 3c and Supplementary Fig. 6c). EXTL3 neutralizing antibody treatment also successfully inhibited REG3B-induced phosphorylation of MEK and ERK in 266-6 and AR42J cell lines and mouse and human primary acinar cells, as well as total KRAS expression in mouse and human primary acini (Fig. 6f with quantification data in Supplementary Fig. 3e–g and Supplementary Fig. 7). Therefore, the ADM process was attributed to the interaction between REG3B and EXTL3 on the acinar cell membrane and subsequent activation of the downstream RAS-RAF-MEK-ERK signaling pathway.

JAK2/STAT3 is another well studied signaling pathway involved in PDAC tumorigenesis and REG3A-related research. Loncle et al.[30] found that REG3B served as a downstream molecule of interleukin-17 during ADM, and further promoted pancreatic cancer cell growth through the GP130/JAK2/STAT3 pathway. Thus, we also explored the impact of REG3B on JAK2/STAT3 signaling in pancreatic acinar cells with TGFα treated cells as positive control. TGFα activated JAK/STAT3 signaling pathway after 72 h in both AR42J and 266-6 cell lines. Remarkably, exogenous REG3B alone failed to activate p-JAK or p-STAT3 after 30 min and 72 h of treatment, in either 266-6 or AR42J cell lines (Fig. 7 and Supplementary Fig. 8). However, REG3B combined with EXTL3 neutralizing antibody enhanced p-JAK2 and p-STAT3 expression (Fig. 7c–e). These data suggest that REG3B binding to EXTL3 may preferentially activate MAPK signaling, rather than JAK2/STAT3.

## Discussion

Initially discovered as an inflammation-induced regenerative protein in pancreas and other organs[31,32], REG3A/REG3B attracted new attention due to its involvement in PDAC development and

progression[16,30,33–36]. In *Reg3B*-deficient mice, Loncle et al.[30] showed that PanIN lesions in the oncogenic *Kras*-driven PDAC model were attenuated and Gironella et al.[33] showed that pancreatic tumor growth was impaired. In the 3D culture of mouse acinar cells, we have previously reported our observation that REG3A promoted ADM formation with concurrent activation of the mitogen-activated protein kinase pathway[21]. This study is designed to further delineate the role of REG3A/REG3B in the formation and maintenance of ADM and its molecular mechanism.

We were unable to detect any REG3A protein or mRNA in normal human pancreatic acini. However, both REG3A protein and mRNA were markedly elevated in ADM tissue adjacent to PDAC. REG3A protein and mRNA were present in the PDAC zone, but the level of expression was lower than that of the ADM zone. This data indicates that REG3A is primarily involved in the ADM process, i.e. the early stage of PDAC tumorigenesis.

We were able to induce ADM formation in both normal human and mouse primary acinar cells by adding recombinant REG3A or REG3B protein, respectively, into the 3D culture media in vitro. We observed a transient and reversible ADM phenotype in the wild type mice with caerulein-induced pancreatitis in the absence of exogenous REG3B. In contrast, *Reg3b* TG Cae or WT Cae+REG3B mice sustained ADM for more than 1 week after the final caerulein injection. Furthermore, PanIN were observed in some TG Cae mice. ADM persisted in WT Cae+ REG3B mice for up to 10 weeks. The combination of both in vitro 3D culture and the dynamic in vivo data convincingly proved that REG3B is involved in the formation and maintenance of persistent ADM. The fact that no ADM was observed in REG3B-treated WT mice without caerulein treatment and *Reg3b* TG mice without caerulein treatment indicates that acinar injury, as a result of inflammation induced by caerulein or trauma inflicted during the isolation of the acinar cells from the pancreas, is the pre-requisite for ADM initiation. Endogenous or exogenous REG3B can only augment the extent and promote persistent ADM after the initial injury occurs.

Mutation of the *Kras* oncogene has been considered to be the critical driver of PDAC tumorigenesis[37]. Most PDAC tumorigenesis studies were conducted in cancer cells and mouse models harboring *Kras* mutations[38,39,40]. The fact that we were able to induce ADM in both in vitro and in vivo models without the need to introduce *Kras* mutation supports the notion, that *Kras* mutation is not an absolutely required molecular event in the ADM process[41,42]. On the other hand, the fact that no invasive PDAC developed in any of our *Reg3b* TG Cae or WT Cae+REG3B mice suggest that additional molecular drivers including mutant *Kras* will be required for the development of full-scale invasive PDAC[43].

The RAS-RAF-MEK-ERK signaling pathway is involved in the initiation and malignant progression of various types of cancers[44,45]. Lineage-tracing studies and other reports demonstrated the

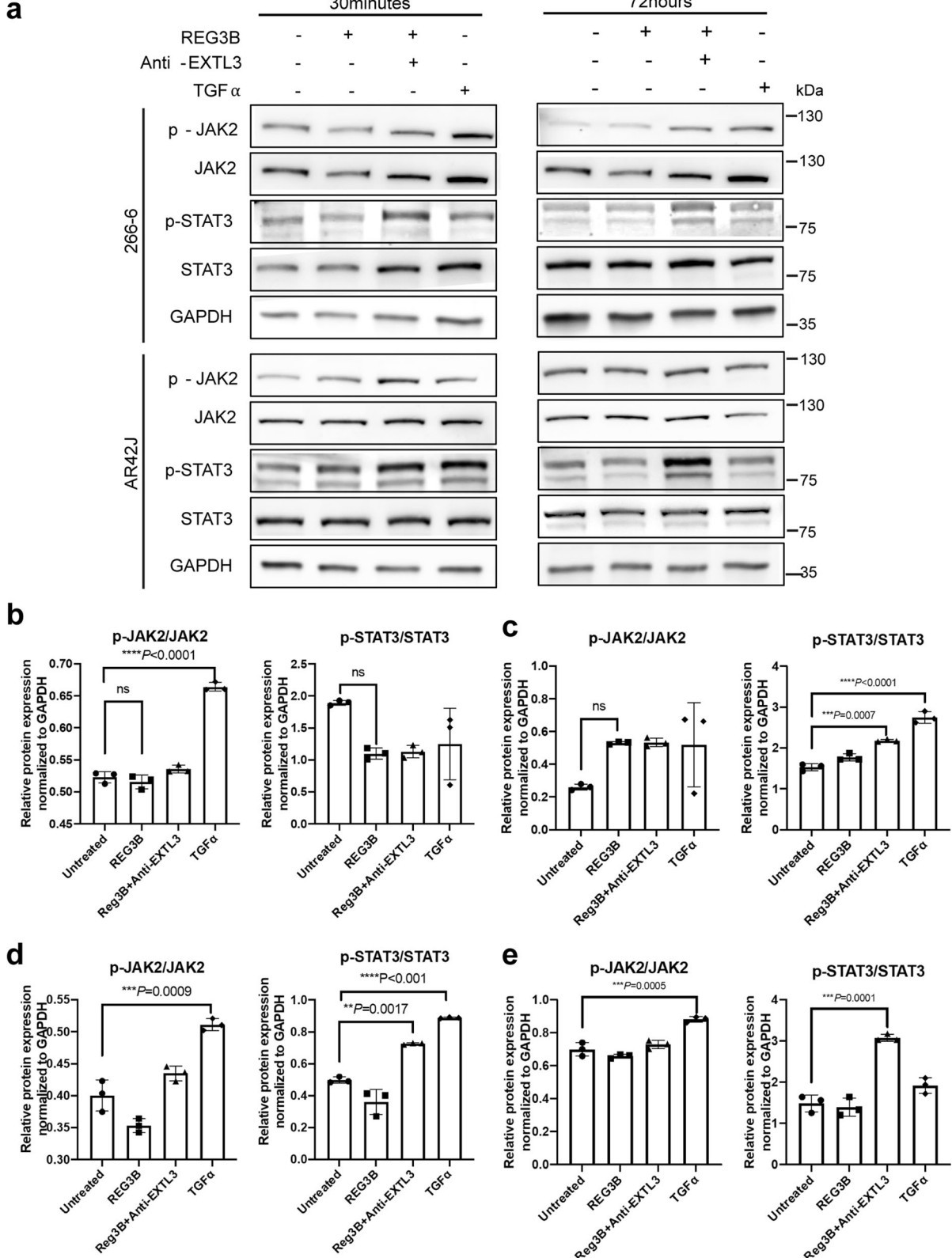

**Fig. 7 JAK2/STAT3 is not involved in REG3B-induced ADM. a** Western blot analysis of related protein expression in 266-6 and AR42J cell lines 30 min and 72 h after different stimulations, with quantification data. **b, c** Expression level of JAK2/STAT3 signaling components in the 266-6 cell line under different conditions for 30 min (**b**) and 72 h (**c**). **d, e** Expression level of JAK2/STAT3 signaling components in the AR42J cell line under different conditions for 30 min (**d**) and 72 h (**e**) ($n = 3$, one-way ANOVA test). REG3B treatment alone did not increase p-JAK nor p-STAT3 expression at two timepoints in two cell lines but did increase their expression once the receptor EXTL3 was blocked.

requirement of RAS-RAF-MEK-ERK signaling in *Kras*-induced ADM[46,47]. Although it has been shown that REG3A, along with other Reg family members, could mediate components of RAS-RAF-MEK-ERK signaling pathway in pancreatic cancer cells[48], pancreatic β cells[49] and gastric cancer cells[50], the detailed mechanism by which REG3A/REG3B regulates pancreatic acinar cells undergoing ADM was poorly described. In vivo, we observed the expression of downstream proteins of the RAS-RAF-MEK-ERK signaling pathway by western blot, and found an increase of active RAS (RAS-GTP) protein in the pancreatic tissue of the WT Cae +REG3B and TG Cae mice. In vitro, REG3A treatment of primary human acinar cells, REG3B treatment of primary murine acinar cells as well as the acinar cell line 266-6, all consistently stimulated the expression of active RAS, KRAS, p-BRAF, p-MEK, and p-ERK. Suppressing the *Reg3b* gene by siRNA dramatically reduced the expression of phosphorylated downstream proteins of the RAS-RAF-MEK-ERK signaling pathway. Treatment with the RAF inhibitor LY3009120 or the MEK1/2 inhibitor Trametinib successfully blocked REG3B-activated p-MEK or p-ERK. In conjunction with decreased protein expression, both inhibitors were able to reverse REG3B-induced ADM in the 3D culture of mouse primary acinar cells at the concentrations required to block REG3B-stimulated activity of these kinases. These approaches from different angles have clearly proven that REG3A/REG3B promotes ADM through the RAS-RAF-MEK-ERK signaling pathway. However, our data by no means indicates that RAS-RAF-MEK-ERK signaling is the exclusive pathway. In fact, Liu et al.[51] reported that REG3B could also activate the GP130-JAK2-STAT3 signaling pathway to accelerate pancreatic cancer cell growth. Future, more extensive studies may unveil other signaling pathways that are also involved in the REG3B-driven ADM process.

For REG3B/REG3A to exert its function, it requires an entry point on the acinar cells, i.e., a receptor to facilitate the interaction with downstream molecules within the cell. Since EXTL3 is more highly expressed on the cell membrane of pancreatic acini than on other cell types and has been implicated as the putative receptor for REG3A in epithelial keratinocytes and for REG1 in neurites[18,52,53], we evaluated whether EXTL3 could interact with REG3B. Dual immunofluorescence staining demonstrated co-localization of REG3B and EXTL3 in REG3B-treated acinar cell lines and in the in vitro 3D culture model of human primary acinar cells. Co-immunoprecipitation experiments revealed an interaction between REG3B and EXTL3. Silencing EXTL3 with a neutralizing antibody in human and mouse primary acinar cells resulted in a reduction of ADM events and inhibition of BRAF, MEK, and ERK phosphorylation in human and mouse primary acinar cells and 266-6 and AR42J cells. This was also shown in 266-6 cells by *Extl3* siRNA treatment. Since other proteins, such as REG1A can also bind to EXTL3 and activate the RAS-RAF-MEK-ERK signaling pathway, silencing EXTL3 may also reduce its downstream signaling that was activated by other proteins. We also screened several other previously reported potential receptors for REG3B such as GP130[30] and EGFR[51], but did not find any positive co-immunoprecipitation with REG3B in acinar cell lines (Supplementary Fig. 3d). The fact that we were unable to validate the result of others could be due to multiple factors, such as differences in the exact experimental conditions. In addition, the targeting receptor for REG3A may vary among different cell types and diseases. In our opinion, negative CO-IP results cannot completely exclude the possibility of other proteins being a potential receptor for REG3B.

It seems that the candidate receptor EXTL3 was not involved in JAK2/STAT3 activation after exogenous REG3B stimulation. In vitro ADM formation induced by REG3B is primarily dependent on downstream RAS-RAF-MEK-ERK signaling through EXTL3. JAK2/STAT3 was only activated when the primary receptor EXTL3 was blocked by a neutralizing antibody. Our data provides evidence that REG3B interacts with EXTL3 on the acinar cell membrane and this interaction promotes ADM by activating the downstream RAS-RAF-MEK-ERK signaling pathway.

Previously, we reported high expression of REG3B in mouse islet cells and its protective role against streptozotocin-induced diabetes mellitus[23]. In this study, we found high protein and mRNA expression of REG3B in both human and mouse pancreatic acinar cells that underwent ADM. In both settings, REG3A/REG3B functions like an "autocrine" molecule, where the protein binds to the receptor on the cell membrane and affects the cell that produces it. The identification of the EXTL3 receptor on the acinar cell membrane provides further supporting evidence for the "autocrine" mechanism of REG3A/REG3B.

In summary, our results clearly demonstrate that, in the absence of *Kras* mutation, REG3A/REG3B can induce and maintain ADM by interacting with EXTL3 and activating the downstream RAS-RAF-MEK-ERK signaling pathway (Fig. 8). In patients with pancreatitis or other types of pancreas injury, it might be worthwhile to measure the REG3A protein level in the pancreatic fluid through endoscopic retrograde cholangiopancreatography (ERCP), or in pancreatic biopsy tissue, or in patient's serum in order to predict the risk of early PDAC tumorigenesis. Targeting REG3A/REG3B, its receptor EXTL3, or other downstream molecules involved in this dynamic process could potentially interrupt the ADM process and prevent early PDAC carcinogenesis.

## Methods

**Human samples, cell lines, and mouse models**. This study was approved by the Institutional Research Ethics Board of the McGill University Health Center (MP-37-2018-4399, MP-37-2018-3171). Normal human pancreatic tissues ($n = 6$) were obtained from the Department of Surgery, McGill University Health Center. Formalin fixed paraffin embedded (FFPE) human PDAC tissue blocks were obtained from the archived tissue at the Department of Pathology, McGill University Health Center. Informed consents were signed by all patients involved in the study.

The pancreatic acinar cell lines AR42J (CRL-1492™) and 266-6 (CRL-2151™) were purchased from ATCC and cultured in a 37 °C and 5% CO₂ incubator. AR42J and 266-6 cells were maintained in RPMI1640 medium (Invitrogen) and Dulbecco's Modified Eagle Medium (DMEM, Wisent), respectively. Both were supplemented with 20% fetal bovine serum, 100 μg/mL penicillin, and 100 μg/mL streptomycin.

Animal protocols were approved by the McGill University Animal Care Committee (AUP 2012-7052). C57BL/6J mice were purchased from the Jackson Laboratory. RIP-I/Reg3B transgenic mice were established and validated by genotyping, as previously described by our group[23]. Adult male C57BL/6J mice (6–12 weeks old) were deprived of food 12 h prior to the in vivo experiments, but given free access to water. Acute pancreatitis was induced by intraperitoneal (i.p.) injection of caerulein (Sigma, C9026) dissolved in PBS at a dose of 50 μg/kg at hourly intervals eight times daily, for two consecutive days[54]. Some mice were sacrificed one day post-caerulein injection (day 3), others were followed by one i.p. injection of recombinant REG3B protein (100 μg/kg) dissolved in PBS daily for seven consecutive days and sacrificed 1 week, 2 weeks, 4 weeks, or 10 weeks later. Recombinant human REG3A and mouse REG3B protein were purchased from R&D Systems (5940-RG and 5110-RG). PBS served as a negative control for injection.

**Primary acinar cell isolation and three-dimensional culture**. Primary pancreatic acinar cells were isolated from the pancreas of six patients (Supplementary Table 1). Briefly, the pancreas was dissected, washed twice with ice-cold HBSS, minced into 1–5 mm pieces and digested with collagenase P (37 °C with a shaker)[46]. The digestion was terminated by an equal volume of ice-cold HBSS with 5% FBS. The digested pancreatic tissue was washed and pipetted through a 500 μm mesh and then a 105 μm mesh. The supernatant was added dropwise to 20 mL HBSS containing 30% FBS. The acinar cells were pelleted (1000 rpm/min for 2 min at 4 °C) and suspended in 10 mL RPMI1640 complete medium (1% FBS, 0.1 mg/mL trypsin inhibitor and 1 μg/mL dexamethasone), referred to as "3D culture media". The cell suspension was gently mixed with an equal volume of matrigel (Corning) and plated in a matrigel pre-coated 8-well chamber (300 μL per well). After solidification, 200 μL of 3D culture media, with or without REG3B (100 nM) or TGFα (50 ng/mL) was added. Media was replaced every other day. On day 5, the amount of ductal structures was counted in all wells

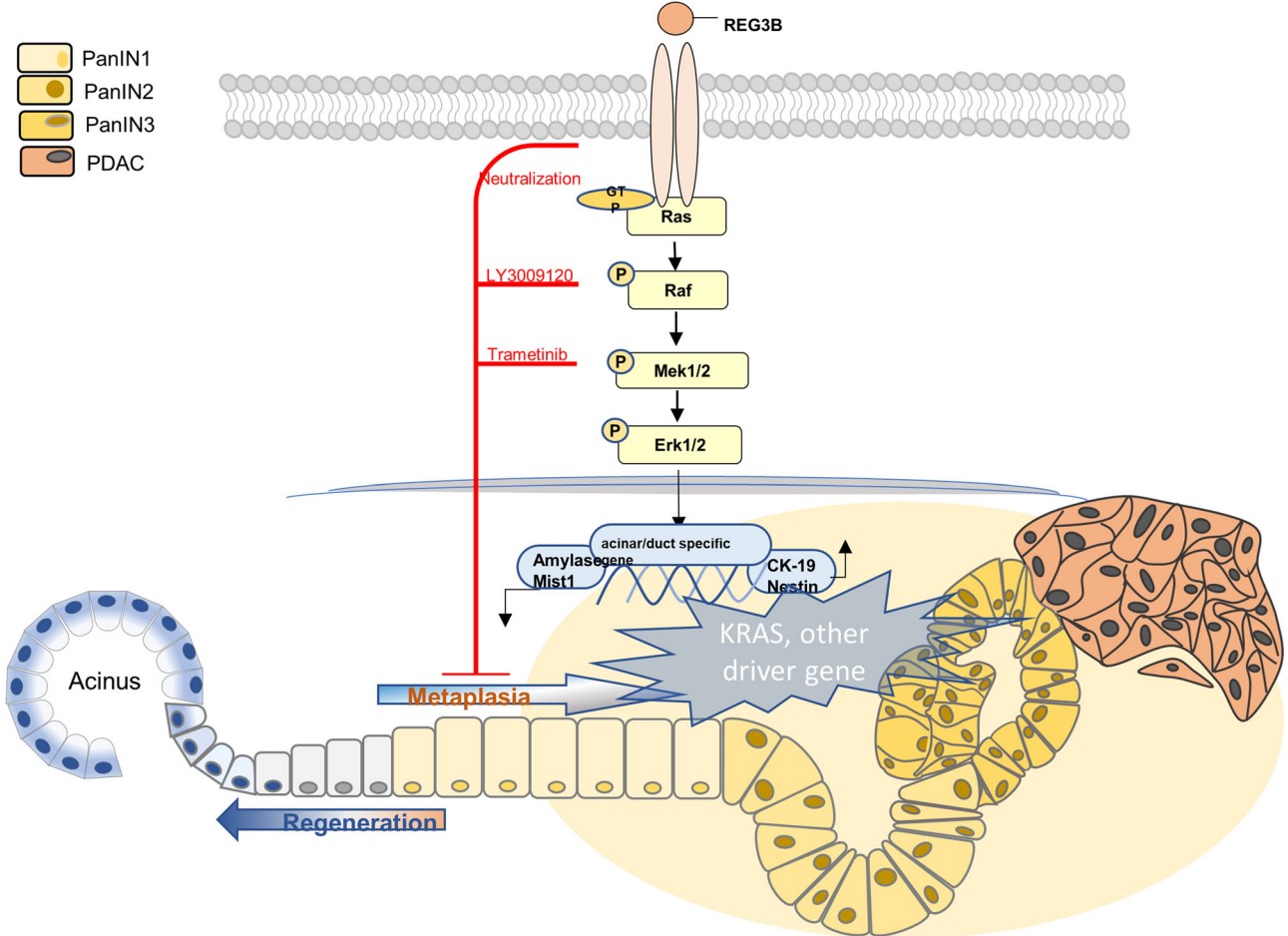

**Fig. 8 Schematic illustration of the molecular mechanism of REG3B/REG3A-driven ADM.** REG3B/REG3A binds to its receptor, EXTL3 receptor on the acinar cell membrane, and promotes ADM by activating the downstream RAS-RAF-MEK-ERK signaling pathway, in the absence of oncogenic Kras mutation. Targeting REG3B/REG3A, neutralizing its receptor EXTL3, or inhibiting downstream signaling molecules, such as B-RAF (LY3009120) or MEK1/2 (Trametinib), could interrupt the ADM process and potentially prevent early PDAC carcinogenesis.

from fifteen 20× fields on the EVOS microscope (Thermo Fisher Scientific) and bright field images were recorded.

**Histology, histochemistry, immunohistochemistry, and immunofluorescence.** Formalin fixed tissue was processed, embedded in paraffin, and cut into 5 μm sections. Hematoxylin and Eosin (H&E) (Thermo Fisher Scientific, 7221, 7111) staining was performed according to the clinical laboratory standard. Immunohistochemistry (IHC) was performed using primary antibodies against REG3A (1:150, R&D Systems, MAB5965), Amylase (1:100, Cell signaling technology) and CK19 (1:100, DSHB). Immunofluorescence staining was performed using primary antibodies against CK19 (1:100, DSHB, TROMAII), Amylase (1:100, Cell signaling technology, 3796S), REG3B (1:100, R&D Systems, AF5110 or AF1996) and EXTL3 (1:70, Biotech, NBP1-31645) according to the manufacturers' instructions.

**Western blotting.** Proteins were extracted, fractionated by SDS-PAGE, and transferred to nitrocellulose membrane (Bio-Rad). Blots were blocked, incubated with primary antibodies (1:1000) at 4 °C overnight, then washed and incubated with secondary antibodies (HRP-conjugated goat anti-mouse, anti-rabbit, and rabbit anti-goat IgG were used, depending on the primary antibodies used). The membranes were briefly incubated with ECL detection reagent (Bio-Rad) to visualize the proteins and images were then captured by ImageQuant Las4000 (GE Healthcare Bio-Science). The primary antibodies were: p-BRAF (T401,Abcam, ab68215), BRAF (Cell Signaling Technology, 14814S), p-MEKk1/2 (Cell Signaling Technology, 9154S), MEK1/2 (Cell Signaling Technology, 4694S), p-ERK1/2 (Cell Signaling Technology,T202/Y204, 4370S), ERK1/2 (Cell Signaling technology, 5013S), GAPDH (Invitrogen, MA5-15738), REG3B (R&D Systems, AF5110, AF1996), EXTL3 (Biotech, NBP1-31645), Amylase (Cell signaling technology, 3796S), p-JAK2 (Cell Signaling technology, 3776), JAK2 (Cell Signaling technology, 3230), p-STAT3 (Cell Signaling technology, 9145), STAT3 (R&D Systems, MAB1799) and KRAS (Biotech, NBP2-45536). Active RAS was measured using the Active Ras Detection Kit (Cell Signaling Technology #8821) according to the

manufacturers' instructions. Full, uncropped blot images are included in Supplementary Figs. 4–8.

**Co-immunoprecipitation (Co-IP) assays.** Total protein lysates of 266-6 and AR42J cell lines and primary acini were obtained from 100 mm dishes at 90% confluence. Normal serum was added to the lysate and kept on ice for 1 h. Two milligram of proteins were pre-cleared by incubation with 50 μL of Protein-G agarose beads (Invitrogen) for 1 h at 4 °C with gentle agitation. An aliquot of supernatant was kept as the input lysate prior to immunoprecipitation. Protein lysates were incubated overnight at 4 °C on a rocking wheel, with anti-REG3B or anti-EXTL3 antibody (20 μg/mL). Then, 50 μL of beads were added to the samples, and the mixture was rotated for 2 h at 4 °C. After three washing steps in lysis buffer to reduce non-specific binding to the beads, the lysate-bead/antibody conjugate mixture (used as after-clearing lysate) was eluted with 50 μL 2× Laemmli sample buffer without DTT for 10 min at 50 °C (elution 1). Fifty microliter of 2× Laemmli sample buffer with DTT (100 mM) was added to the pelleted beads and boiled for 5 min (elution 2). Input lysate, after-clearing lysate, elution 1 and elution 2 were analyzed by western blotting for EXTL3 or REG3B.

**RNA isolation and real-time RT-qPCR.** Following the manufacturer's instructions, total RNA was extracted from mouse pancreas or 266-6 cell line with the RNeasy Plus Mini Kit (Qiagen). Single-stranded cDNA was then synthesized from 1 μg of extracted mRNA using a cDNA synthesis kit (SuperScript IV, Thermo Fisher Scientific). DNA from formalin-fixed paraffin embedded (FFPE) tissues was extracted with the RecoverAll Total Nucleic Acid Isolation Kit (Ambion). Real-time PCR was performed on an Applied Biosystems 7500 Sequence Detection system with SYBR Green Master Mix (Applied Biosystems) containing 1/10 of total volume cDNA and 10 nmol of each pair of primers (sequences shown in Supplementary Table. S2). The cycling conditions were: 50 °C for 2 min, 95 °C for 2 min for initial denaturation; 95 °C for 15 s, 57 °C for 15 s, and 72 °C for 1 min for 40 cycles followed by the melting curve stage. The relative expression of target gene

was evaluated based on the threshold cycle (Ct), and calculated as $2^{-\Delta\Delta CT}$. All genes were normalized to the internal control YWHAZ (Tyrosine 3-Mono-oxygenase/Tryptophan 5-Monooxygenase Activation Protein Zeta) or RPL13A (Ribosomal Protein L13a). All samples were tested in at least three independent experiments.

**RNA interference and transient transfection**. Small interfering RNA (siRNA) oligonucleotides targeting *Extl3* and *Reg3b* (Supplementary Table. S3) were designed and synthesized by Origene. 266-6 and AR42J cell lines were transfected with siRNA (20 nM) or scrambled siRNA according to the manufacturer's instructions. Cells were grown in a 6-well plate and transfected with siRNA at 200 pmol diluted in RNAimax and Optimem medium (Invitrogen). Knockdown efficiency of *Reg3b* or *Extl3* was confirmed by western blotting.

**Statistics and reproducibility**. Each experiment was performed at least three independent times with at least duplicate samples. Data were expressed as mean ± SD. A one-way ANOVA or Student's *t*-test was used between two groups to compare the difference by SPSS 21.0, and formatted with GraphPad Prism version 6.0 software. Tukey multiple-comparison posttest was performed for statistical analysis of multiple groups. A two-tailed *P* value of <0.05 was considered as statistically significant. *$P < 0.05$; **$P < 0.01$; ***$P < 0.001$. ns no significance.

**Reporting summary**. Further information on research design is available in the Nature Research Reporting Summary linked to this article.

## Data availability
The datasets generated and/or analyzed during the current study can be found in Supplementary Data. 1 or are available from the corresponding author upon request.

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

## Acknowledgements

We thank Fazila Chouiali (Histopathology Platform, Research Institute of the McGill University Health Center), Shi Bo Feng and Min Fu (Molecular Imaging Platform, Research Institute of the McGill University Health Center) for their technical support. This work was supported by Canadian Institute of Health Research (J.L.L., Z.H.G.; PJ9-173593 and PJT-175208), and China Scholarship Council (HRZ; 201606280269).

## Author contributions

Conceptualization, Z.H.G., A.L.G.C., H.R.Z., and Q.L.; Methodology, H.R.Z., N.K., J.M.P., and A.L.G.C.; Software, H.R.Z., J.M.P., and A.L.G.C.; Validation, A.L.G.C., J.M.P., and H.R.Z.; Formal analysis, A.L.G.C., J.M.P., and H.R.Z.; Investigation, H.R.Z., J.M.P., N.K., and Q.L.; Resources, G.Z. and Y.W.; Original draft preparation, H.R.Z.; Review and editing, Z.H.G., A.L.G.C., J.M.P., and H.R.Z.; Supervision, Z.H.G., B.Y.S., J. L.L., and A.G.

## Competing interests

The authors declare no competing interests.
