## [Transparent Peer Review File · Communications Biology]

Reviewers' comments:

Reviewer #1 (Remarks to the Author):

This paper from Zhang et. al. investigates the role of hReg3A/Reg3 β in promoting pancreatic acinar to ductal metaplasia (ADM). The authors investigate the underlying mechanism and identify the EXTL3 receptor and MAPK signaling pathway as downstream mediators. The study is novel and of interest to the community. Pancreatic cancer continues to have a very poor prognosis and insights into its early precursor lesions are extremely important. The study makes remarkable use of multiple model systems and supports its conclusions using several orthogonal assays. The design and methodology are commendable.

My concerns and comments are outlined below.

1. The paper demonstrates the involvement of Reg3 β in persistent ADM. However, its role in the generation of PanIN has not been explored in detail. The only mention is the presence of focal PanIN in TG Cae mice (page 5). The authors correctly point out in their discussion (page 8) that none of the mice developed invasive PDAC, which probably needs additional drivers. Conclusions such as "Reg3 β is involved in malignant progression of ADM" are thus far reaching. I would encourage revising any such text that appears multiple times through all sections in the paper as well as figure 7.
2. How often did TG Cae mice develop PanIN? Were any PanINs observed with WT Cae + recombinant Reg3 β ?
3. The expression of Reg3 β alone in mouse and human acinar culture models was sufficient to induce ADM. However, TG mice did not develop ADM and required Cae. How do the authors interpret this given the role of inflammation in pancreatic transformation?
4. To avoid confusion, references to the mouse and human gene homologs should be made using standardized gene nomenclature guidelines. Similarly, genes and proteins should be distinguished using standard nomenclature guidelines
<https://www.genenames.org/about/guidelines/>
http://www.informatics.jax.org/mgihome/nomen/short_gene.shtml
5. In figure 6G: si-EXTL3 alone reduces signaling regardless of Reg3 β . How do the authors interpret this result? Considering the potential off-target effects of siRNAs, more than one construct could be used to validate the findings.
6. The authors mentioned that they screened for other receptors in their co-immunoprecipitation experiments. This data should be included. How do the authors interpret the discrepancy between previous findings and this study?
7. The extent of MAPK pathway activation with Reg3 β appears variable across figures 3-6. Please include the number of repeat experiments for all western blotting and imaging results in the legends. Confocal images could benefit from the quantification of multiple fields and replicates as has been done in figure 6D.
8. IHC results in figure 1 could be provided for all patients given the small sample size. If qPCR (supplementary figure 1A) was conducted on PDAC tissue, these results should be included.
9. Were human pancreatic acinar cells isolated from multiple patients or represent replicates of the same original organoid?
10. I'm curious as to why different cell lines were used to investigate the effects of LY3009120 and Trametinib. Was TGF beta included as a positive control for the human acini?
11. Introduction: Please remove the last line "the comprehensive data..." or discuss in detail in the discussion. Can a reference be provided for the homology between the mouse and human genes?

Reviewer #2 (Remarks to the Author):

Summary of the paper

Pancreatic adenocarcinoma accounted for over 90% of all pancreatic cancer and have poor prognosis, mainly due to late diagnosis and poor treatment outcome. Thus more understanding of PDAC initiation is important for early detection and developing preventative therapy. ADM is one feature that has strong correlation with PDAC initiation but not fully understood. In this paper, Dr. Zhang et al used multiple models to identify the promoting role of hReg3A/mReg3 β for ADM formation, and provided a working mechanism by activating RAS-RAF-MEK-ERK pathway. One of the paper highlights is including both mouse and human models to reflect potential species difference and extend to clinical relevance. Overall impression

The paper is well constructed, and conclusions were fairly supported by evidences provided. However, there are some concerns listed as follows that need to be addressed to strengthen the paper.

First of all, authors should distinguish further the intracellular and extracellular functions of hReg3A/mReg3 β . Most of the results seemed to point to extracellular function, such as the injection hReg3A and addition of hReg3A/mReg3 β , including function of binding to EXTL3 to activate RAS-RAF-MEK-ERK pathway; however some evidence provided such as the protein and mRNA expression of hReg3A/mReg3 β increase in cells lead to similar phenotype pointed to intracellular function and doesn't really go along with the working mechanism that hReg3A/mReg3 β binds with EXTL3 to function.

Secondly, as previously characterized in cerulean induced pancreatitis model, persistent inflammation plays critical role in mediating ADM, and further transition to PanIn. The authors didn't explore how significant a role that hReg3A/mReg3 β plays on regulating JAK2/STAT3 signaling pathway, and how it might have affected ADM in this paper.

Lastly, authors should provide more quantification of in vitro 3D culture system. 3D system is a powerful system, but due to the innate property that both acinar cells and ductal cells will grow out of the primary samples there will be some baseline differences of each population's proportion that varies from batch to batch. For each assay or culture, authors should use the taken images to quantify the base line % of acinar (amylase+) and % of ductal cell (CDK19+) in untreated samples to show the effect of hReg3A/mReg3 β on acinar transition to ductal cells more convincingly.

Comment on revision

- 1) For Figure 1A, can author use another ADM marker in addition to H&E characterization to show co-localization of Reg3A expression with ADM area? How many patient samples did authors stain with IHC and see a similar effect? The microenvironment of PDAC (high inflammation and high KRas activity) could cause ADM similar phenotype, but the mechanism to trigger this tumor adjacent ADM might not be similar to the pre-lesion ADM which occurs before PDAC onset. Can author comment on how relatable are these two different stages of ADM?
- 2) Figure S1A, can author also provide RT-PCR result of other ADM markers (Mist1 and CK19) for the same sample to confirm identity of normal vs. ADM tissues?
- 3) Figure 2F, GAPDH band is not clear, especially for Caerulein treated samples.
- 4) Can author provide IHC staining of Reg3 β on 1) Caerulein treated samples to see if it colocalize with ADM area; 2) on day 7 post caerulein when no ADM phenotype present?
- 5) it is worth to point out that Reg3 β is specifically expressed by islet cells instead of in acinar or ductal cells in the transgenic model, which might support more of an extracellular function of Reg3 β .
- 6) Can author quantify the area% of ADM in 2,4 and 10 weeks after caerulein treatment between Reg3 β treated and WT mice?
- 7) Co-IP result is sub-optimal, and need a better blot. EXTL3 band for 266-6 IB:EXTL3 is a blur not a band. Please also provide explanation of Elution1 and Elution2 in figure legend,.
- 8) in Figure 6A, does colocalization of hReg3A/mReg3 β with EXTL3 in AR42J and 166-6 cause transition of acinar to ductal in 2D culture? Is there any ADM phenotype if authors treat AR42J and

166-6 with hReg3A/mReg3 β in 3D culture? Would author please add TNF α as positive control to see if it will cause increase of ADM? Can author show staining in mouse ADM in addition to human ADM?

Reviewer #1 (Remarks to the Author):

This paper from Zhang et. al. investigates the role of hReg3A/Reg3 β in promoting pancreatic acinar to ductal metaplasia (ADM). The authors investigate the underlying mechanism and identify the EXTL3 receptor and MAPK signaling pathway as downstream mediators. The study is novel and of interest to the community. Pancreatic cancer continues to have a very poor prognosis and insights into its early precursor lesions are extremely important. The study makes remarkable use of multiple model systems and supports its conclusions using several orthogonal assays. The design and methodology are commendable.

My concerns and comments are outlined below.

1. **Comment:** The paper demonstrates the involvement of Reg3 β in persistent ADM. However, its role in the generation of PanIN has not been explored in detail. The only mention is the presence of focal PanIN in TG Cae mice (page 5). The authors correctly point out in their discussion (page 8) that none of the mice developed invasive PDAC, which probably needs additional drivers. Conclusions such as “Reg3 β is involved in malignant progression of ADM” are thus far reaching. I would encourage revising any such text that appears multiple times through all sections in the paper as well as figure 7.

Answer: Thank you for pointing this out. We have removed the wording “malignant progression” throughout the text and added other possible drivers between persistent ADM and PanIN in Figure 8. Our conclusion has changed to “Reg3B is involved in ADM formation and maintenance” (lines 34, 79, 99-100, 137-138, 195, 225, 231, 236, 239-240, 301, 311, etc.).

2. **Comment:** How often did TG Cae mice develop PanIN? Were any PanINs observed with WT Cae + recombinant Reg3 β ?

Answer: In the TG Cae group, 2 out of 5 mice (40%) developed PanIN at D10. In the WT Cae + recombinant REG3B treated mice group, 0 out of 5 mice (0%) mice PanIN at D10. This information has been added to the text lines 29-31, 126-127, 364-365.

3. **Comment:** The expression of Reg3 β alone in mouse and human acinar culture models was sufficient to induce ADM. However, TG mice did not develop ADM and required Cae. How do the authors interpret this given the role of inflammation in pancreatic transformation?

Answer: We believe that acinar injury, as a result of inflammation induced by caerulein or trauma during the isolation of the acinar cells from the pancreas, is the pre-requisite for ADM initiation. REG3B, endogenous or exogenous, can only augment the extent and promote persistent ADM after the injuries occur.

The following findings supported our explanation: (1) In caerulein treated wild type mice without REG3B treatment, there was transient ADM (Figure 3b and 3e); (2) In the 3D culture model without REG3B treatment, there was small number of ADM (Figure 4a and 4b); (3) No ADM was seen in Reg3 β TG mice without Cerulean treatment (Figure 3b).

We have added this important point in the discussion line 240-245.

4. **Comment:** To avoid confusion, references to the mouse and human gene homologs should be made using standardized gene nomenclature guidelines. Similarly, genes and proteins should be distinguished using standard nomenclature guidelines

<https://www.genenames.org/about/guidelines/>

http://www.informatics.jax.org/mgihome/nomen/short_gene.shtml

Answer: We have double checked our manuscript to make sure we are following the standard nomenclature guidelines.

5. **Comment:** In figure 6G: si-EXTL3 alone reduces signaling regardless of Reg3 β . How do the authors interpret this result?

Answer: Other proteins such as Reg1a could also bind to EXTL-3 and activate the RAS-RAF-MEK-ERK signaling pathways. Silencing EXTL3 could reduce its downstream signaling that was activated by other proteins as well.

Comment: Considering the potential off-target effects of siRNAs, more than one construct could be used to validate the findings.

Answer: We received 3 constructs of si-EXTL3. Only one out the three had an impact on the MEK-ERK signaling. This information has been added to the results section lines 198-199.

6. **Comment:** The authors mentioned that they screened for other receptors in their co-immunoprecipitation experiments. This data should be included.

Answer: The co-IP data of other receptors has been added as supplementary Fig.2j.

Comment: How do the authors interpret the discrepancy between previous findings and this study?

Answer: The fact that we were unable to validate others' results could be due to multiple factors such as differences in the exact experimental conditions and the low sensitivity of western blotting. Sometimes, weak binding of REG3B with other receptors could be washed out during manipulation of cell samples. In our opinion, negative CO-IP results can not completely exclude the possibility of other protein being a potential receptor for REG3B. We have added this comment in the discussion lines 292-295.

7. **Comment:** The extent of MAPK pathway activation with Reg3 β appears variable across figures 3-6. Please include the number of repeat experiments for all western blotting and imaging results in the legends. Confocal images could benefit from the quantification of multiple fields and replicates as has been done in figure 6D.

Answer: Thank you for the suggestion. The numbers of repeated experiments for all western blotting and imaging results have been added to the related figure legends. Please see new figure legends.

8. **Comment:** IHC results in figure 1 could be provided for all patients given the small sample size.

Answer: Thank you for your suggestion. IHC results for all patients have been provided in supplementary Fig.1. Unless the reviewer feels strongly, we prefer to use the original Figure 1 as it appears less crowded and more clearly demonstrates our findings.

Comment: If qPCR (supplementary figure 1A) was conducted on PDAC tissue, these results should be included.

Answer: The qPCR data on PDAC tissue has been added to supplementary Fig.2a.

9. **Comment:** Were human pancreatic acinar cells isolated from multiple patients or represent replicates of the same original organoid?

Answer: Human pancreatic acinar cells were isolated from 4 patients and tested in 4 separate experiments. This information has been added to the manuscript line 475. Details of those 4 patients are provided in the supplementary Table 1.

10. **Comment:** I'm curious as to why different cell lines were used to investigate the effects of LY3009120 and Trametinib.

Answer: We use two different cells acinar cells lines, one from mouse and one from rat, to cross validate our results.

Comment: Was TGF beta included as a positive control for the human acini?

Answer: TGF beta was not used as positive control for the culture of human primary acinar cells in our study.

11. **Comment:** Introduction: Please remove the last line “the comprehensive data...” or discuss in detail in the discussion.

Answer: Thanks for the suggestion. We have removed the sentence “the comprehensive data...” in the introduction.

Comment: Can a reference be provided for the homology between the mouse and human genes.

Answer: Thank you for the suggestion. A reference has been added specifically addressing the homology between the mouse and human Reg genes. Please see line 639.

Loncle C, Bonjoch L, Folch-Puy E, Lopez-Millan MB, Lac S, Molejon MI, et al. REG3 β Plays a Key Role in IL17RA Protumoral Effect—Response. *Cancer Res* 2016;76(7):2051.

Reviewer #2 (Remarks to the Author):

Summary of the paper

Pancreatic adenocarcinoma accounted for over 90% of all pancreatic cancer and have poor prognosis, mainly due to late diagnosis and poor treatment outcome. Thus more understanding of

PDAC initiation is important for early detection and developing preventative therapy. ADM is one feature that has strong correlation with PDAC initiation but not fully understood. In this paper, Dr. Zhang et al used multiple models to identify the promoting role of hReg3A/mReg3 β for ADM formation, and provided a working mechanism by activating RAS-RAF-MEK-ERK pathway. One of the paper highlights is including both mouse and human models to reflect potential species difference and extend to clinical relevance.

Overall impression

The paper is well constructed, and conclusions were fairly supported by evidences provided. However, there are some concerns listed as follows that need to be addressed to strengthen the paper.

1. **Comment:** First of all, authors should distinguish further the intracellular and extracellular functions of hReg3A/mReg3 β . Most of the results seemed to point to extracellular function, such as the injection hReg3A and addition of hReg3A/mReg3 β , including function of binding to EXTL3 to activate RAS-RAF-MEK-ERK pathway; however some evidence provided such as the protein and mRNA expression of hReg3A/mReg3 β increase in cells lead to similar phenotype pointed to intracellular function and doesn't really go along with the working mechanism that hReg3A/mReg3 β binds with EXTL3 to function.

Answer: Thank you for pointing this out. In our opinion, hREG3A/mREG3B functions like an "autocrine" molecule. The intracellular protein and mRNA expression of hReg3A/mReg3 β indicates that the hREG3A/mREG3B protein is produced within the pancreatic acinar cells. However, in order for hREG3A/mREG3B to exert its function, it requires it being secreted to the extracellular environment and binding to EXTL3 on the acinar cell membrane.

We have added this point in the discussion lines 306-309.

2. **Comment:** Secondly, as previously characterized in cerulean induced pancreatitis model, persistent inflammation plays critical role in mediating ADM, and further transition to PanIn. The authors didn't explore how significant a role that hReg3A/mReg3 β plays on regulating JAK2/STAT3 signaling pathway, and how it might have affected ADM in this paper. [New experiment is suggested on (1) how Reg3A regulate JAK2/STAT3, and (2) how that may have affected ADM?]

Answer: Our experiments clearly demonstrated that the JAK2/STAT3 pathway is not activated by REG3 β . The result of our experiments on JAK2/STAT3 signaling pathway has been added in the result section lines 207-215 and Figure 7. A sentence summarizing this result has been added to the discussion section lines 296-299.

3. **Comment:** Lastly, authors should provide more quantification of in vitro 3D culture system. 3D system is a powerful system, but due to the innate property that both acinar cells and ductal cells will grow out of the primary samples there will be some baseline differences of each population's proportion that varies from batch to batch. For each assay or culture, authors should use the taken images to quantify the base line % of acinar (amylase+) and % of ductal cell (CDK19+) in untreated

samples to show the effect of hReg3A/mReg3 β on acinar transition to ductal cells more convincingly. [Ask for better quantification and better/negative control]

Answer: Thank you for the suggestion. The quantification data of the 3D images has been added to supplementary Fig.2i. These quantification data further supported our conclusion.

Comment on revision

1) **Comment:** For Figure 1A, can author use another ADM marker in addition to H&E characterization to show co-localization of Reg3A expression with ADM area?

Answer: We have performed REG3A and CK19 double staining on all human samples and provided the data in the new supplementary Fig.1.

Comment: How many patient samples did authors stain with IHC and see a similar effect?

Answer: Immunohistochemistry was performed on 9 human samples. The image of one patient was in Figure 1. The images of other 8 patients were included in the supplementary Fig.1.

Comment: The microenvironment of PDAC (high inflammation and high KRas activity) could cause ADM similar phenotype, but the mechanism to trigger this tumor adjacent ADM might not be similar to the pre-lesion ADM which occurs before PDAC onset. Can author comment on how relatable are these two different stages of ADM?

Answer: ADM, whether it is pre-lesion or caused by the microenvironment of PDAC, is a premalignant condition and has the potential to progress to PDAC with additional drivers. In the setting of PDAC microenvironment-induced, it may contribute to the local expansion of tumor.

2) **Comment:** Figure S1A, can author also provide RT-PCR result of other ADM markers (Mist1 and CK19) for the same sample to confirm identity of normal vs. ADM tissues?

Answer: We have provided the RT-PCR results of other ADM markers in the revised supplementary Fig.2a.

3) **Comment:** Figure 2F, GAPDH band is not clear, especially for Caerulein treated samples.

Answer: We have repeated the experiment and replaced the Figure 2f with a better image.

4) **Comment:** Can author provide IHC staining of Reg3 β on 1) Caerulein treated samples to see if it colocalize with ADM area; 2) on day 7 post caerulein when no ADM phenotype present?

Answer: 1) We have added the IHC staining images on Caerulein treated samples to the new Figure 2, which co-localized with the ADM area. 2) The REG3B IHC was negative on day 7 post caerulein when no ADM phenotype was present. Since the result is negative, we have decided not to include it in the figure.

5) **Comment:** it is worth to point out that Reg3 β is specifically expressed by islet cells instead of in acinar or ductal cells in the transgenic model, which might support more of an extracellular function of Reg3 β .

Answer: Thank you for the suggestion. We have added this point in the discussion Lines 303 to 309.

6) **Comment:** Can author quantify the area% of ADM in 2, 4 and 10 weeks after caerulein treatment between Reg3 β treated and WT mice?

Answer: The quantification of the percentage areas of ADM in 2, 4, 10 weeks after caerulein treatment between REG3B treated and WT mice have been added to supplementary Fig.2c. The quantification results further supported our conclusion.

7) **Comment:** Co-IP result is sub-optimal, and need a better blot. EXTL3 band for 266-6 IB:EXTL3 is a blur not a band. Please also provide explanation of Elution1 and Elution2 in figure legend,

Answer: Thank you for pointing this out.

We have repeated our Co-IP experiment and provided additional data in the revised Figure 6, which further supported our conclusion.

We have provided explanation of Elution 1 and Elution 2 in the methods (lines 524-525) and in the figure legend.

8) **Comment:** In Figure 6A, does colocalization of hReg3A/mReg3 β with EXTL3 in AR42J and 166-6 cause transition of acinar to ductal in 2D culture?

Answer: Although adding REG3B to the culture of AR42J and 266-6 cell lines could activate the molecular cascade in the RAS-RAF-MEK-ERK pathway (Figure 5 B, Figure 6 G-H), we were unable to induce complete ADM phenotype. It seems that other factors (such as adhesion molecules, stromal regulators, etc.) in the primary pancreatic acini also contributed to the complete formation of ADM phenotype.

Comment: Is there any ADM phenotype if authors treat AR42J and 166-6 with hReg3A/mReg3 β in 3D culture?

Answer: After multiple attempts, we were unable to induce complete ADM phenotype by adding REG3B or TGF α in both 2D and 3D culture of AR42J and 266-6 cell lines. To our knowledge, all the published literature always uses primary acinar cells and there is no successful ADM model using culture of acinar cell lines.

Comment: Would author please add TNF α as positive control to see if it will cause increase of ADM?

Answer: We didn't use TNF α as a positive control. We use TGF α and we were unable to induce ADM phenotype by adding TGF α to the 3D culture of human acinar cells (data not shown). Our finding is consistent with previous reports by Liu J, et al.

Liu, J., Akanuma, N., Liu, C. *et al.* TGF- β 1 promotes acinar to ductal metaplasia of human pancreatic acinar cells. *Sci Rep* **6**, 30904 (2016). <https://doi.org/10.1038/srep30904>

Comment: Can author show staining in mouse ADM in addition to human ADM?

Answer: We performed REG3B immunohistochemistry on mouse ADM. The results were consistent with what we have seen in human samples. These results have been provided in Figure 2h.

Reviewers' comments:

Reviewer #1 (Remarks to the Author):

Zhang et.al. have revised their manuscript considerably. My comments are as follows:

1. In the section "Reg3 β induces ADM through the RAS-RAF-MEK-ERK signaling pathway": Description of results in Figure 5A relating to TG mice is missing in the text. Also, please add a short description for LY300912 and Trametinib eg. RAF and MEK inhibitor in the text.
2. Fig. 6: Co-IP experiments: Please add an explanation for Elution 1 and 2 to the legend.
3. In the revision, the authors have included results for IP with other receptors gp130 and EGFR (supplementary 2J). This figure citation and results section is missing in the text. The discussion (line 825) says data not shown for these experiments.
4. Figure 7A: Please add that TGF alpha is a positive control in the text.
5. I fully recognize the time and effort involved in conducting additional experiments. I do not wish to be inconsiderate, but I am bothered by the fact that additional siRNA constructs for EXTL3 did not impact signaling. One way to address this could be repeating the experiment in 6H using neutralizing antibody in the second cell line as well as human acinar cell lines. Also, for clarification: is the neutralizing antibody construct the same as the detection antibody? This was unclear from the methods. If the antibody works to neutralize human EXTL3, inclusion of human acinar cells in Figure 6C would be an added bonus.

Reviewer #2 (Remarks to the Author):

Authors had addressed most of comments and provided stronger supporting evidences.

One minor issue need to be addressed is as follows:

- 1) supplementary Fig.1. i-l and m-p white balance is off, please adjust as done in supplementary Fig.1.q.

Response to Reviewers' comments

Reviewer #1 (Remarks to the Author):

Zhang et.al. have revised their manuscript considerably. My comments are as follows:

1. **Comment:** In the section "Reg3 β induces ADM through the RAS-RAF-MEK-ERK signaling pathway": Description of results in Figure 5A relating to TG mice is missing in the text. Also, please add a short description for LY300912 and Trametinib eg. RAF and MEK inhibitor in the text.

Response: Thank you for pointing this out.

We have added to the text a description of results in Figure 5A relating to TG mice, please see page 6, lines 163 to 164.

We have added to the text a short description for LY300912 and Trametinib eg. RAF and MEK inhibitor, please see lines page 6, 171 to 172.

2. **Comment:** Fig. 6: Co-IP experiments: Please add an explanation for Elution 1 and 2 to the legend.

Response: Thank you for pointing this out. We have added to the legend an explanation for Elution 1 and 2. Please see page 14, lines 432-433

3. **Comment:** In the revision, the authors have included results for IP with other receptors gp130 and EGFR (supplementary 2J). This figure citation and results section is missing in the text. The discussion (line 825) says data not shown for these experiments.

Response: Thank you for pointing this out. The results for IP with other receptors gp130 and EGFR is now in supplementary Figure 3d. We have cited this result in the discussion section, please see page 10, lines 300-306. We have removed the phrase "data not shown" in the discussion.

4. **Comment:** Figure 7A: Please add that TGF alpha is a positive control in the text.

Response: Thank you for pointing this out. We have added TGF alpha is a positive control in the text please see lines 218-221.

5. **Comment:** I fully recognize the time and effort involved in conducting additional experiments. I do not wish to be inconsiderate, but I am bothered by the fact that additional siRNA constructs for EXTL3 did not impact signaling. One way to address this could be repeating the experiment in 6H using neutralizing antibody in the second cell line as well as human acinar cell lines.

Response: Thank you for the suggestion. We have performed additional experiments and presented the data regarding the effect of neutralizing antibody on human primary acinar cells, mouse primary acinar cells, 266-6 and AR42J cell lines. Please see the result section lines 207-210, and discussion lines 295-298 (Figure 6f, and supplementary Figure 3e-g).

Reviewer #2 (Remarks to the Author):

Authors had addressed most of comments and provided stronger supporting evidences.

One minor issue need to be addressed is as follows:

1) **Comment:** supplementary Fig.1. i-l and m-p white balance is off, please adjust as done in supplementary Fig.1.q.

Response: Thank you for the suggestion. We have adjusted the white balance in Fig1 i-l and m-p.

REVIEWERS' COMMENTS:

Reviewer #1 (Remarks to the Author):

My concerns have been adequately addressed by the authors.